# Criteria and Non-Criteria Antiphospholipid Antibodies and Cancer in Patients with Involuntary Weight Loss

**DOI:** 10.3390/jpm13111549

**Published:** 2023-10-29

**Authors:** Simona Caraiola, Laura Voicu, Anda Baicus, Cristian Baicus

**Affiliations:** 1Fifth Department-Internal Medicine (Cardiology, Gastroenterology, Hepatology, Rheumatology, Geriatrics), Family Medicine, Occupational Medicine, Faculty of Medicine, “Carol Davila” University of Medicine and Pharmacy, 050474 Bucharest, Romania; 2Internal Medicine Department, Colentina Clinical Hospital, 020125 Bucharest, Romania; 3Laboratory Department, The University Emergency Hospital, 050098 Bucharest, Romania

**Keywords:** antiphospholipid antibodies, cancer, involuntary weight loss

## Abstract

Cancer patients have higher prevalences of antiphospholipid antibodies (aPLs), occasionally associated with thrombotic events. A cross-sectional study regarding the presence of criteria (IgG/IgM anti-cardiolipin-aCL, anti-β2 glycoprotein I-aβ2GPI) and non-criteria (IgG/IgM anti-phosphatidylserine-aPS, anti-phosphatidylethanolamine-aPE, anti-prothrombin-aPT) aPLs in 146 patients with involuntary weight loss was performed. None of the patients had thrombotic events during the study. Out of the 36 cancer patients, 33 had non-hematologic malignancies. In the cancer subgroup, 60% of the patients had at least one positive aPL, with significantly more patients being positive for aβ2GPI IgG compared with the non-cancer subgroup—*p* = 0.03, OR = 2.23 (1.02–4.88). When evaluating the titres, aCL IgG/IgM, aβ2GPI IgG, aPE IgG, and aPS IgG had significantly higher values in cancer patients, the best cancer predictor being aβ2GPI IgG—AUC 0.642 (0.542–0.742). Gastrointestinal cancer patients were studied separately, and aCL IgM positivity was significantly higher—*p* = 0.008, OR = 6.69 (1.35–33.02). Both the titres of aCL IgM (*p* = 0.006) and aPS IgM (*p* = 0.03) were higher in the gastrointestinal cancer subgroup, with aCL IgM being the best predictor for gastrointestinal cancer development—AUC 0.808 (0.685–0.932). Despite criteria and non-criteria aPLs being frequent in cancer, their connection with thrombosis in these patients is probably dependent on other important risk factors and needs further research.

## 1. Introduction

Antiphospholipid antibodies (aPLs) are autoantibodies mainly involved in the pathophysiology of the antiphospholipid syndrome (APS) [1]. The aPLs that can be found in the classification criteria of APS—namely the lupus anticoagulant (LAC), the anticardiolipin antibodies (aCL), and the anti-β2 glycoprotein I antibodies (aβ2GPI)—are generally known as “criteria aPLs” [2]. However, many other aPLs—such as anti-prothrombin antibodies (aPT), anti-phosphatidylethanolamine antibodies (aPE), and anti-phosphatidylserine antibodies (aPS)—have been described and are grouped under the name of “non-criteria aPLs”. Their role in the pathogenic mechanisms and diagnosis of APS is currently under debate [3].

The global incidence of malignancies has been remarkably increasing in the last years, and the expected number of cases in the next decades is far from negligible [4].

According to the available data, malignancy is responsible for 30% of the involuntary weight loss cases, while non-malignant or psychosocial disorders are other important possible etiologies [5,6,7]. In contrast, the percentage of cancer patients who experience unintentional weight loss can rise to over 80% in certain cancer types [8]. As unexplained weight reduction might represent a timely red flag for malignancy, taking into consideration a possible cancer diagnosis when evaluating these patients is essential [9].

It became a well-known fact that cancer patients have a higher frequency of thromboses when compared with the general population [10]. In addition, thrombosis, especially under the form of venous thromboembolism, represents one of the most important causes of mortality in patients diagnosed with malignancy [11]. The incidence of thrombotic events seems to be higher shortly before and after the diagnosis [12,13]. Pancreatic, lung, gastric, ovarian, or brain cancers seem to be associated with a higher risk of thrombosis than other cancer types. Increased age, comorbidities, or chemotherapy are found among the risk factors cited in the literature [12]. Radiotherapy, hormonal therapy, surgery, immobility, and the presence of central venous catheters are also recognized as thrombosis-risk-enhancing factors [13,14]. Dehydration caused by vomiting or diarrhea during chemotherapy, as well as blood transfusions, may supplementarily increase the risk of thrombosis in cancer patients.

Recent findings suggest a higher prevalence of aPLs in patients with malignancies, when compared with patients without cancer [15]. Although aPL positivity in malignancies can elevate the risk of thrombosis, their presence might as well remain asymptomatic. Increased levels of aPLs have been found in both hematologic and non-hematologic various types of cancer [15,16]. It has been shown that gastrointestinal cancer patients have a fivefold rise of aCL prevalence when compared with healthy controls. Some studies also highlight a significant increase in the presence of aCL in genitourinary or lung cancer patients. In regard to the presence of aβ2GPI or LAC, no statistically significant differences were detected between patients with diverse types of solid cancers and healthy controls [17]. Non-criteria aPLs seem to have a higher prevalence in uterine cancer patients compared to patients with non-malignant gynecological pathology. Therefore, aPL positivity could conceivably represent an augmentation factor for thrombosis in cancer [18]. Despite these findings, the utility of routine testing for aPLs in cancer patients still remains uncertain [16].

The topic of aPL production mechanisms in cancer is waiting to be elucidated by future research. Some hypotheses have so far been formulated. According to these, cancerous cells might be directly secreting aCL or monoclonal immunoglobulins with aCL or LAC activity. Another theory is that aPLs could be generated by the immune system as a reaction to tumor antigens [19]. Recent studies have highlighted the increased phospholipid synthesis by malignant cells, in order to sustain their accelerated cellular replication. These phospholipids represent targets for the aPLs [20]. Moreover, arguments have been formulated in support of a bidirectional relationship, suggesting that the presence of aPLs in cancer patients is not only a consequence of the tumoral process, but might also be involved in neoplastic growth. aPLs seem to augment tumoral angiogenesis and the progression of malignancy [19], and they might be a cancer risk factor, especially for hematologic malignancies [21].

According to the latest recommendations, in cancer patients who develop venous acute thromboses, the anticoagulant therapy is preferentially performed with non-vitamin K antagonist oral anticoagulants (NOAC) [22]. However, in cancer patients that associate APS, as in all patients diagnosed with APS, vitamin K antagonists (VKA) are the preferred anticoagulant therapy [23]. Therefore, in patients with malignancies associated with thrombosis, especially in the case of recurrence, testing for aPLs might be useful in the establishment of the therapeutic plan.

The aim of this research was to evaluate the presence of criteria and non-criteria aPLs in a group of patients presenting with involuntary weight loss of unknown etiology. Moreover, the study intended to compare the aPL profiles of patients who were eventually diagnosed with cancer to the ones of the patients with non-malignant diagnoses.

## 2. Materials and Methods

The conducted study had a cross-sectional design. Consecutive patients admitted into the Internal Medicine Department of Colentina Clinical Hospital for involuntary weight loss of unknown etiology were prospectively enrolled.

Only patients 18 years old or older were recruited. An unintentional decrease of at least 5% of body weight in the six previous months was a mandatory inclusion criterium. For the patients in whom the exact amount of lost weight could not be documented, a 5-point Likert scale (“very much”, “much”, “average”, “little”, “not at all”) was used to estimate weight reduction. Only patients declaring to be “very much” or “much” affected were included. In addition, the weight loss estimated by the Likert scale had to be confirmed either by a clothing size change or by the reiteration of a relative. Patients with voluntary body weight decrease were excluded, along with patients who had a known diagnosis leading to weight loss.

The participating physicians were allowed to establish a personalized investigation plan for every patient. A six-month follow-up face-to-face meeting or phone call was implemented, in order to verify the occurrence of thrombosis, the establishment of the diagnosis, the survival, and any other significant health changes. A total of 24 deaths were registered during this time, while 49 patients were lost to follow-up. None of the included patients had a personal history of thrombotic events, and none of them developed thrombosis in the six-month follow-up period.

All the patients gave their informed consent before being enrolled. All the data were collected and analyzed under the assurance of anonymity. The study protocol was approved by the Ethics Committee of Colentina University Hospital Bucharest.

For every patient, blood samples were collected at the time of enrolment. All the samples were centrifuged and stored at −70 °C. A single determination of aPLs was performed for each of the study samples. The panel included IgG and IgM isotypes of aCL, aβ2GPI, aPT, aPE, and aPS. Logistical limitations of the study protocol did not allow testing for LAC.

ELISA kits (Aesku Diagnostics, Wendelsheim, Germany), and a Chemwell 2910 Analyzer (Awareness Technology, Palm City, FL, USA) were used to determine the titres of aPLs. A value of 18 U/mL or more was considered positive, while values under 12 U/mL was deemed as negative. The values ranging between these extremes were registered as equivocal.

The methodology was more fully described in references [24,25,26].

Continuous variables with parametric distribution were reported as frequencies, and as mean ± standard deviation. The means were compared with Student’s *t*-test. Variables not normally distributed were analyzed with the Mann–Whitney test, Wilcoxon test, and Kendall rank correlation coefficient (r). The degrees of correlation have been defined as follows: 0.00–0.10, no correlation; 0.11–0.30, low correlation; 0.31–0.50, medium correlation; 0.51–0.70, high correlation; 0.71–1.00, very high correlation. When comparing categorical variables, the Chi-square test was used. Odds ratios (OR) and 95% confidence intervals (CI) were determined. Receiver operating characteristic curves (ROC) were performed, and areas under the curve (AUC) were measured. The two-tailed hypothesis testing was used, and a *p*-value less than 0.05 was considered statistically significant. The statistical analysis was conducted by using SPSS 20.0 software (IBM Corporation, Armonk, NY, USA).

## 3. Results

A total of 146 patients were enrolled in the study. The main demographic characteristics of the study group are presented in Table 1. The percentages of males and females proved to be almost equal (52% versus 48%). The mean age at enrolment in the entire population was 64 years. The enrolment age was found to be statistically significantly higher in cancer patients than in those without cancer (*p* = 0.004). No significant difference was found between the cancer and non-cancer subgroups regarding gender (*p* = ns). About 25% (36) of the patients were eventually diagnosed with different types of cancer. Among them, three patients had hematologic cancers—two myelomas, and one lymphoma. The other 33 patients were diagnosed with non-hematologic cancers. This subgroup comprised five patients with colon cancer, two with gastric cancer, two with pancreatic cancer, one with biliary cancer, two patients with liver cancer, and two cases of metastatic liver lesions with unknown primary site. In addition, seven pulmonary cancers, two pleural cancers, five renal cancers, two bladder cancers, two ovarian cancers, and one sarcoma were registered. The rest of them had various non-neoplastic diagnoses, such as collagen vascular diseases, psychiatric disorders, cardiovascular, pulmonary, digestive, or neurologic pathologies (Table 2).

In the analyzed population, 61.11% (22) of the cancer patients were positive for at least one of the tested aPLs, while patients without cancer had a positivity rate of 46.36% (51). However, the registered difference was not statistically significant (*p* = 0.12). As shown in Table 3, the most frequent positive aPLs encountered in the entire study group were aβ2GPI IgM (30%), aPT IgM (18%), and aPE IgM (17%). For the cancer subgroup, the ranking is maintained, 44% of the cancer patients being positive for aβ2GPI IgM, 19% for aPT IgM, 19% for aPE IgM, and 19% for aCL IgM. The number of aβ2GPI IgM positive cancer patients was significantly higher when compared with the non-cancer subgroup (*p* = 0.03). A *p* value near to the significance level (*p* = 0.08) was registered for aCL IgM.

Significantly higher titres of aCL IgG (*p* = 0.03), aCL IgM (*p* = 0.01), aβ2GPI IgG (*p* = 0.001), aPE IgG (*p* = 0.02), and aPS IgG (*p* = 0.02) were detected in the cancer subgroup when compared with patients without cancer (Table 4). For aPE IgM, a near-to-significance level *p* value was found (*p* = 0.08).

The comparative evaluation of aPL titres and positivity revealed no significant differences between the hematologic and non-hematologic cancer patients. 

When performing ROC analysis for evaluating the utility of criteria and non-criteria aPLs in predicting cancer development (Table 5), aβ2GPI IgG registered the largest AUC, with a value of 0.642 (0.542–0.742), closely followed by aCL IgM—AUC 0.633 (0.529–0.738), and aPE IgG—AUC 0.623 (0.518–0.728).

Out of the 22 patients who proved to have a double aPL positivity profile, only one was positive for both of the analyzed criteria aPLs (aCL and aβ2GPI), while the remaining 21 patients were positive for two non-criteria or one of the criteria and one of the non-criteria aPLs. No statistically significant differences were registered between the patients with or without cancer regarding multiple positive aPLs profiles (Table 6), although for double positivity the value of *p* was situated close to the significance limit (*p* = 0.06).

The aPL profiles of gastrointestinal cancer (colon cancer and gastric cancer) patients were analyzed separately (Table 7). The aCL IgM positivity was significantly higher in these patients, compared with the rest of the study population (*p* = 0.008). The difference is maintained when analyzing the titres—*p* = 0.006 for aCL IgM (Table 8). Moreover, the titres of aPS IgM are significantly increased in gastrointestinal cancer patients (*p* = 0.03), although the number of positive aPS IgM patients was not significantly different in patients with or without gastrointestinal cancer.

The ROC analysis pointed towards aCL IgM as the best predictor for gastrointestinal cancer development in the studied population—AUC 0.808 (0.685–0.932). The second highest AUC was registered for aPS IgM—0.735 (0.539–0.932). The values of AUC for the other tested aPLs are presented in Table 9.

In addition, the positivity and titres of aPLs in pulmonary and renal cancer were analyzed independently. No statistically significant differences were obtained between these patients and the rest of the study group.

The analyzed correlations between criteria and non-criteria aPLs in the study group, although statistically significant in most of the cases, generally proved to be of weak or medium degree (Table 10). The aPS registered the highest coefficient values—aPS IgG-aCL IgG r = 0.72 (*p* < 0.01), aPS IgM-aCL IgM r = 0.58 (*p* < 0.01). No correlation was found between aPT IgG and aCL IgM r = 0.11 (*p* = ns), while between aPT IgG and aCL IgG, a weak correlation was detected—r = 0.19 (*p* < 0.01).

When analyzing the correlations of criteria and non-criteria aPLs in the subgroup of cancer patients (Table 11), no correlations were found between aCL IgM and the IgG and IgM isotypes of aPT (r = −0.09, r = 0.10 respectively, *p* = ns), as well as in the aPS IgG-aβ2GPI IgM (r = 0.13, *p* = ns), aPS IgM-aβ2GPI IgG (r = 0.16, *p* = ns), aPS IgG-aβ2GPI IgG (r = 0.17, *p* = ns), and aPS IgM-aCL IgG (r = 0.19, *p* = ns) pairs. Contrarily, the highest correlation coefficients were registered between aPS IgM and aCL IgM (r = 0.73, *p* < 0.01), and between aPS IgG and aCL IgG (r = 0.64, *p* < 0.01). Despite their lack of correlation with aCL, aPT correlated with aβ2GPI under low or moderate degree (aPT IgG-aβ2GPI IgG r = 0.31, *p* < 0.01, aPT IgG-aβ2GPI IgM r = 0.26, *p* < 0.05, aPT IgM-aβ2GPI IgG r = 0.41, *p* < 0.01, aPT IgM-aβ2GPI IgM r = 0.41, *p* < 0.01).

## 4. Discussion

It is estimated that about 5% of the healthy young adults are aPL carriers. The percentage rises with age and chronic diseases, reaching 50% in elderly individuals with comorbidities [17]. However, only a minority of the aPL-positive subjects will turn out to have APS [16]. The results of previous studies indicate the variability of aPL-carrier cancer patients’ evolution. In a study evaluating the association of aPLs with malignancy, 21% of the patients met the APS Sapporo criteria, while thrombotic manifestations were registered for 71% of the study sample [27]. In the study population analyzed by Yoon et al., 87% of the cancer patients developed venous thrombosis, and arterial thrombosis was found in 24% of them [28]. Contrarily, Vassalo et al. registered venous thrombotic manifestations in only 4% of their cancer patients; none of the participants turned out to have APS [29]. In a 137 cancer patient sample, Bazzan et al. found only nine patients with venous thromboembolism, while the rest remained free of thrombotic events. Only 5 patients finally met the APS classification criteria [30].

The link between aPLs and thrombosis in cancer patients could not yet be incontestably demonstrated [18]. According to Reinstein et al., the pathogenicity of aPLs cannot be sustained by the detected levels, despite their elevation in patients with different types of malignancies [31]. A recent study found no statistically significant differences regarding the incidence of thrombosis in lymphoma patients with positive aPLs when compared to patients with lymphoma, but without aPLs [32]. Vassalo et al. have shown that aPLs do not associate with thrombosis or medium-term survival in critically ill cancer patients, and that they generally have a transient nature [29]. The transient aPL positivity in solid malignancy patients has also been shown by Font et al., who suggest the absence of aPLs’ pathogenic role in thromboses [33]. Moreover, the results of Gómez-Puerta et al. indicated that 35% of the evaluated cancer aPL-positive patients became negative after following oncologic treatment [34]. Some authors suggest that not all aPL isotypes are associated with thrombosis, and that transient IgM aPLs do not have any pathogenic role in thrombotic events [21].

Contrarily, recent research suggests that the presence of aPLs increases the risk of thrombosis in cancer patients [19]. Thromboembolic events may represent the clinical debut of underlying malignancy [27]. Their prevalence has been found to vary between 16 and 23% in aPL-positive cancer patients [17].

None of the patients evaluated in our study had a history of thrombosis. Furthermore, none of them developed thrombosis during the hospital admission, and none of the patients who were available for the six-month follow-up visit/phone call had a thrombotic event in this time. Similarly, Miesbach et al. followed non-Hodgkin lymphoma patients with high levels of aCL IgM for at least two years, with no thromboembolic events being reported [35]. One study monitored the drop in aCL levels after surgery or chemotherapy. The patients were followed for the next 12 months, and none of them developed thrombosis [36]. As our patients only had a unique determination of aPLs, the transient or persistent nature of these antibodies could not be appreciated.

For the frequency of aPLs in cancer patients, values between 1.4% and 74% were reported [15]. In our study population, more than 60% of the cancer patients turned out to be positive for at least one of the determined aPLs. However, the percentage was not significantly superior to the one registered in the non-cancer subgroup (46%). No other studies analyzing the prevalence of aPLs in involuntary weight loss patients could be found. 

In the present research, the most prevalent aPLs in cancer patients were aβ2GPI IgM (44%), followed by aPT IgM (19%), aPE IgM (19%), and aCL IgM (19%). The only significant difference discovered in our research between cancer and non-cancer patients in terms of aPL positivity was registered for aβ2GPI IgM, which was significantly more prevalent in the cancer subgroup—*p* = 0.03, OR 2.23 (1.02–4.88). For aCL IgM, a near-to-significance value was noted (*p* = 0.08). The unquestionable dominance of IgM isotypes in our thrombosis-free study population might represent a supplementary argument for their lack of pathogenicity in cancer patients.

Bazzan et al. analyzed the titres of aPLs in cancer patients. They found that even though the laboratory positivity criteria were not met, low aPLs titres had a six-times higher prevalence in cancer patients, when compared with the control group. Notwithstanding these findings, no significant differences regarding thrombosis-free survival could be found between the two groups [30]. Medium or high aPL titres proved to have a more important clinical significance [18].

Consistent with the previous mentioned results, significantly higher values were observed for aCL IgG (*p* = 0.03), aCL IgM (*p* = 0.01), aβ2GPI IgG (*p* = 0.001), aPE IgG (*p* = 0.02), and aPS IgG (*p* = 0.02) titres in our cancer patients. When analyzing the utility of criteria and non-criteria aPLs in predicting cancer development, aβ2GPI IgG proved to be the best predictor, with an AUC of 0.642 (0.542–0.742). aCL IgM came second—AUC 0.633 (0.529–0.738)—closely followed by aPE IgG—AUC 0.623 (0.518–0.728).

An increased risk of developing aPLs has been found for almost every studied solid cancer subtype. Most of the studies have evaluated the presence of aCL, aβ2GPI, and LAC [15]. In one of the few studies that analyzed the prevalence of non-criteria aPLs in cancer patients, Islam et al. found a high prevalence of these antibodies in both hematologic and solid malignancies [19]. Non-criteria aPLs were found to be more frequent in uterine cancer than in non-malignant gynecologic pathology, while no differences were found for criteria aPLs [37].

In our research, no significant differences were found between hematologic and non-hematologic cancer patients regarding aPL titres or positivity. The most prevalent subtypes of solid malignancies in the study group were gastrointestinal, pulmonary, and renal cancers. In consequence, these categories were analyzed separately. For the last two, no significant differences were found in terms of aPLs titres or positivity, when compared to the rest of the study population. The prevalence of aCL IgM positivity proved to be significantly increased in gastrointestinal cancers—*p* = 0.008, OR 6.69 (1.35–33.02). Moreover, the titres of aCL IgM and aPS IgM were significantly higher in these patients (*p* = 0.006 and *p* = 0.03 respectively). aCl IgM also turned out to be the best predicting aPL for gastrointestinal cancers, with an AUC of 0.808 (0.685–0.932), while aPS IgM registered the second highest value—0.735 (0.539–0.932).

As mentioned before, other studies have found associations between aCL and cancer. Our results were concordant to the ones of previous research regarding the connection of aCL with gastrointestinal cancers [17]. High prevalences of aCL were also described in pulmonary cancer patients [38], though, in this aspect, the present study did not obtain similar findings.

A multiple-aPL-positivity profile has been presented in the literature as a supplementary risk factor for thrombosis in cancer patients [34]. In a recent analysis, Kansuttiviwat et al. showed that double aPL positivity increased the risk for venous thromboembolic events from 3.6 times—which was the value obtained for cancer and single aPLs positivity profile—to 7.4 times [15]. Data linking multiple aPL positivity to cancer are scarce. Our results indicated no significant differences between cancer and non-cancer patients when analyzing multiple aPL positivity profiles, even though, for double positivity, a borderline significance level was found (*p* = 0.06). Most of the double-aPL–positive participants had a combination of one of the criteria and one of the non-criteria aPLs, or two non-criteria aPLs.

The correlations of criteria and non-criteria aPLs were mainly of weak or medium degree. When analyzing the entire study population, the highest correlation coefficients were found between aPS and aCL, the values varying from medium to very high degree of correlation (aPS IgG-aCL IgG r = 0.72, *p* < 0.01). Low or medium correlations were registered between aPS and aβ2GPI. No correlations (aPT IgG and aCL IgM r = 0.11, *p* = ns) or weak correlations were found between aPT and both aCL and aβ2GPI.

In the subgroup of cancer patients, despite the high degree of correlation found in the aPS IgM-aCL IgM (r = 0.73, *p* < 0.01), and aPS IgG-aCL IgG (r = 0.64, *p* < 0.01) pairs, no correlations or low correlations were detected between aPS and the rest of the tested criteria aPLs. No correlations were found between aPT and aCL, except for the aPT IgM-aCL IgG pair that correlated under a low degree (r = 0.25, *p* < 0.05). Low or moderate degrees of correlation were registered between aPT and aβ2GPI.

The relatively low degree of correlation found between aPT and the criteria aPLs is consistent to other results in the literature that had similar findings in a study population of APS patients [39].

The present research has several limitations. First, this study had a cross-sectional design, and only one determination of aPLs was performed. Second, the relatively low number of cancer patients that were included, and the reduced number of aPL-positive patients, might generate a statistical bias. Third, the fact that LAC could not be tested might represent another important disadvantage.

Despite the reduced study population, an extensive panel of aPLs were searched, generating statistically significant results. Moreover, not only the criteria, but also the non-criteria, aPLs were determined. As far as the authors are aware, this study is the first one investigating the prevalence of aPLs in patients with involuntary weight loss, as well as one of the few studies describing the correlations between the criteria and the non-criteria aPLs.

In conclusion, despite the increasing evidence linking criteria and non-criteria aPLs to cancer, the exact implications remain unclear. The involvement of aPLs in thrombotic events seems to vary with their isotype, as well as their transient or persistent character. Furthermore, the connection of aPLs with thrombosis in these patients is most probably influenced by the association of other risk factors. Even though there are still not enough arguments for routinely testing aPLs in cancer patients, future research of the topic might impose their screening, especially in patients with supplementary risk factors for thrombosis. 

## Figures and Tables

**Table 1 jpm-13-01549-t001:** Demographic characteristics of the study population.

		Overall	Cancer	Without Cancer	*p*-Values	OR (95% CI)
Number of Patients		146	36 (24.65%)	110 (75.34%)		
Gender	Male	76 (52.05%)	19	57	0.53	0.03 (0.48–2.20)
	Female	70 (47.95%)	17	53		
Mean age at enrolment (years)		64.34 ± 13.34	69.81 ± 8.65	62.55 ± 14.13	0.004	

**Table 2 jpm-13-01549-t002:** Causes of weight loss in the study population.

Diagnosis	Number of Patients (*n* = 146)
Hematologic cancer	3 (2.05%)
Non-hematologic cancer	33 (22.61%)
Collagen vascular diseases	13 (8.91%)
Acute infection	8 (5.48%)
Cardiovascular pathology	3 (2.05%)
Pulmonary pathology	2 (1.37%)
Digestive pathology	28 (19.18%)
Dibetes mellitus	6 (4.11%)
Biermer anemia	2 (1.37%)
Thyroid pathology	3 (2.05%)
Saturnism	2 (1.37%)
Neurologic pathology	3 (2.05%)
Psychiatric disorder	28 (19.18%)
Unknown diagnosis	12 (8.22%)

**Table 3 jpm-13-01549-t003:** Serological characteristics of the study population.

aPLs	Overall (Positive/Negative) (*n* = 146)	Cancer (Positive/Negative) (*n* = 36)	Without Cancer (Positive/Negative) (*n* = 110)	*p*-Values	OR (95% CI)
aCL IgG	5/141 (3.42%)	1/35 (2.77%)	4/106 (3.63%)	0.64	0.75 (0.08–7.00)
aCL IgM	17/129 (11.64%)	7/29 (19.44%)	10/100 (9.09%)	0.08	2.41 (0.84–6.90)
aβ2GPI IgG	9/137 (6.16%)	4/32 (11.11%)	5/105 (4.54%)	0.15	2.62 (0.66–10.36)
aβ2GPI IgM	45/101 (30.82%)	16/20 (44.44%)	29/81 (26.36%)	0.03	2.23 (1.02–4.88)
aPT IgG	13/133 (8.90%)	5/31 (13.88%)	8/102 (7.27%)	0.18	2.05 (0.62–6.74)
aPT IgM	26/120 (17.80%)	7/29 (19.44%)	19/91 (17.27%)	0.47	1.15 (0.44–3.02)
aPE IgG	8/138 (5.48%)	2/34 5.55(%)	6/104 (5.45%)	0.63	1.02 (0.19–5.29)
aPE IgM	25/121 (17.12%)	7/29 (19.44%)	18/92 (16.36%)	0.42	1.23 (0.46–3.24)
aPS IgG	7/139 (4.79%)	2/34 (5.55%)	5/105 (4.54%)	0.55	1.23 (0.22–6.66)
aPS IgM	10/136 (6.85%)	3/33 (8.33%)	7/103 (6.36%)	0.46	1.33 (0.32–5.46)

aPLs—antiphospholipid antibodies; aCL—anti-cardiolipin antibodies; aβ2GPI—anti-β2 glycoprotein I antibodies; aPE—anti-phosphatidylethanolamine antibodies; aPS—anti-phosphatidylserine antibodies; aPT—anti-prothrombin antibodies.

**Table 4 jpm-13-01549-t004:** Titres of the tested aPLs in the study population.

aPLs	Overall Median Values (Min–Max) (*n* = 146)	Cancer Median Values (Min–Max) (*n* = 36)	Without Cancer Median Values (Min–Max) (*n* = 110)	*p*-Values
aCL IgG	3 (1–210) U/mL	4 (1–18) U/mL	3 (1–210) U/mL	0.03
aCL IgM	3 (0–300) U/mL	5.5 (0–60) U/mL	3 (0–300) U/mL	0.01
aβ2GPI IgG	5 (0–300) U/mL	5.5 (2–300) U/mL	4 (0–299) U/mL	0.001
aβ2GPI IgM	11 (0–300) U/mL	12.5 (0–151) U/mL	11 (0–300) U/mL	0.12
aPT IgG	5 (0–289) U/mL	6 (0–289) U/mL	5 (0–82) U/mL	0.17
aPT IgM	3.5 (0–300) U/mL	5 (0–60) U/mL	3 (0–300) U/mL	0.48
aPE IgG	3 (0–300) U/mL	4 (0–37) U/mL	2 (0–300) U/mL	0.02
aPE IgM	6 (0–300) U/mL	10 (0–300) U/mL	5 (0–300) U/mL	0.08
aPS IgG	4 (1–237) U/mL	5 (1–20) U/mL	3 (1–237) U/mL	0.02
aPS IgM	5 (0–300) U/mL	6.5 (0–171) U/mL	5 (0–300) U/mL	0.15

aPLs—antiphospholipid antibodies; aCL—anti-cardiolipin antibodies; aβ2GPI—anti-β2 glycoprotein I antibodies; aPE—anti-phosphatidylethanolamine antibodies; aPS—anti-phosphatidylserine antibodies; aPT—anti-prothrombin antibodies.

**Table 5 jpm-13-01549-t005:** Results of ROC analysis evaluating the utility of criteria and non-criteria aPLs in predict-ing cancer development.

aPLs	Area Under the Curve	95% CI	
		Lower Limit	Upper Limit
aCL IgG	0.614	0.515	0.712
aCL IgM	0.633	0.529	0.738
aβ2GPI IgG	0.642	0.542	0.742
aβ2GPI IgM	0.585	0.478	0.693
aPT IgG	0.576	0.461	0.690
aPT IgM	0.538	0.430	0.647
aPS IgG	0.624	0.526	0.723
aPS IgM	0.578	0.470	0.687
aPE IgG	0.623	0.518	0.728
aPE IgM	0.597	0.493	0.700

aPLs—antiphospholipid antibodies; aCL—anti-cardiolipin antibodies; aβ2GPI—anti-β2 glycoprotein I antibodies; aPE—anti-phosphatidylethanolamine antibodies; aPS—anti-phosphatidylserine antibodies; aPT—anti-prothrombin antibodies.

**Table 6 jpm-13-01549-t006:** Multiple aPLs positivity in the study group and subgroups.

	Overall (Positive/Negative) (*n* = 146)	Cancer (Positive/Negative) (*n* = 36)	Without Cancer (Positive/Negative) (*n* = 110)	*p*-Values	OR (95% CI)
Double positivity	22/124 (15.07%)	9/27 (25%)	13/97 (11.81%)	0.06	2.48 (0.96–6.43)
Triple positivity	10/136 (7.53%)	0/36	10/100 (9.09%)	0.12	0.13 (0.007–2.29)
Quadruple positivity	6/140 (4.10%)	2/34 (5.55%)	4/106 (3.63%)	0.63	1.55 (0.27–8.88)
Quintuple positivity	3/143 (2.05%)	2/34 (5.55%)	1/109 (0.90%)	0.15	6.41 (0.56–72.91)

**Table 7 jpm-13-01549-t007:** Serological characteristics of the gastrointestinal cancer study subgroup.

aPLs	Gastrointestinal Cancer (Positive/Negative) (*n* = 7)	Without Gastrointestinal Cancer (Positive/Negative) (*n* = 139)	*p*-Values	OR (95% CI)
aCL IgG	0/7	5/134	0.74	1.63 (0.08–32.32)
aCL IgM	¾	14/125	0.008	6.69 (1.35–33.02)
aβ2GPI IgG	1/6	8/131	0.36	2.72 (0.29–25.48)
aβ2GPI IgM	2/5	43/96	0.89	0.89 (0.16–4.79)
aPT IgG	0/7	13/126	0.75	0.62 (0.03–11.55)
aPT IgM	1/6	25/114	0.80	0.76 (0.08–6.59)
aPE IgG	0/7	8/131	0.98	1.03 (0.05–19.61)
aPE IgM	1/6	24/115	0.83	0.79 (0.09–6.94)
aPS IgG	0/7	7/132	0.96	0.92 (0.04–17.54)
aPS IgM	1/6	9/130	0.42	2.40 (0.26–22.21)

aPLs—antiphospholipid antibodies; aCL—anti-cardiolipin antibodies; aβ2GPI—anti-β2 glycoprotein I antibodies; aPE—anti-phosphatidylethanolamine antibodies; aPS—anti-phosphatidylserine antibodies; aPT—anti-prothrombin antibodies.

**Table 8 jpm-13-01549-t008:** Titres of the tested aPLs in the gastrointestinal cancer study subgroup.

aPLs	Gastrointestinal Cancer Median Values (Min–Max) (*n* = 7)	Without Gastrointestinal Cancer Median Values (Min–Max) (*n* = 139)	*p*-Values
aCL IgG	4 (2–9) U/mL	3 (1–210) U/mL	0.44
aCL IgM	13 (4–25) U/mL	3 (0–300) U/mL	0.006
aβ2GPI IgG	7 (3–34) U/mL	5 (0–300) U/mL	0.21
aβ2GPI IgM	11 (6–151) U/mL	11 (0–300) U/mL	0.64
aPT IgG	6 (1–12) U/mL	5 (0–289) U/mL	0.92
aPT IgM	2 (0–31) U/mL	4 (0–300) U/mL	0.82
aPE IgG	10 (6–96) U/mL	3 (0–300) U/mL	0.11
aPE IgM	6 (2–9) U/mL	6 (0–300) U/mL	0.09
aPS IgG	6 (4–7) U/mL	4 (1–239) U/mL	0.12
aPS IgM	13 (2–22) U/mL	5 (0–300) U/mL	0.03

aPLs—antiphospholipid antibodies; aCL—anti-cardiolipin antibodies; aβ2GPI—anti-β2 glycoprotein I antibodies; aPE—anti-phosphatidylethanolamine antibodies; aPS—anti-phosphatidylserine antibodies; aPT—anti-prothrombin antibodies.

**Table 9 jpm-13-01549-t009:** Results of ROC analysis evaluating the utility of criteria and non-criteria aPLs in predicting gastrointestinal cancer development.

aPLs	Area Under the Curve	95% CI	
		Lower Limit	Upper Limit
aCL IgG	0.585	0.410	0.759
aCL IgM	0.808	0.685	0.932
aβ2GPI IgG	0.637	0.418	0.857
aβ2GPI IgM	0.552	0.379	0.726
aPT IgG	0.510	0.294	0.727
aPT IgM	0.525	0.336	0.714
aPS IgG	0.672	0.561	0.783
aPS IgM	0.735	0.539	0.932
aPE IgG	0.674	0.516	0.831
aPE IgM	0.686	0.566	0.805

aPLs—antiphospholipid antibodies; aCL—anti-cardiolipin antibodies; aβ2GPI—anti-β2 glycoprotein I antibodies; aPE—anti-phosphatidylethanolamine antibodies; aPS—anti-phosphatidylserine antibodies; aPT—anti-prothrombin antibodies.

**Table 10 jpm-13-01549-t010:** The aPLs’ correlations in the study group.

	aCL IgG	aCL IgM	aβ2GPI IgG	aβ2GPI IgM	aPT IgG	aPT IgM	aPE IgG	aPE IgM	aPS IgG	aPS IgM
aCL IgG	1	0.43 **	0.36 **	0.24 **	0.19 **	0.25 **	0.48 **	0.34 **	0.72 **	0.33 **
aCL IgM	0.43 **	1	0.25 **	0.34 **	0.11	0.27 **	0.30 **	0.38 **	0.35 **	0.58 **
aβ2GPI IgG	0.36 **	0.25 **	1	0.40 **	0.26 **	0.25 **	0.50 **	0.32 **	0.40 **	0.22 **
aβ2GPI IgM	0.24 **	0.34 **	0.40 **	1	0.23 **	0.30 **	0.36 **	0.40 **	0.18 **	0.42 **
aPT IgG	0.19 **	0.11	0.26 **	0.23 **	1	0.34 **	0.30 **	0.16 **	0.12 *	0.10
aPT IgM	0.25 **	0.27 **	0.25 **	0.30 **	0.34 **	1	0.28 **	0.33 **	0.18 **	0.30 **
aPE IgG	0.48 **	0.30 **	0.50 **	0.36 **	0.30 **	0.28 **	1	0.38 **	0.47 **	0.32 **
aPE IgM	0.34 **	0.38 **	0.32 **	0.40 **	0.16 **	0.33 **	0.38 **	1	0.29 **	0.46 **
aPS IgG	0.72 **	0.35 **	0.40 **	0.18 **	0.12 *	0.18 **	0.47 **	0.29 **	1	0.30 **
aPS IgM	0.33 **	0.58 **	0.22 **	0.42 **	0.10	0.30 **	0.32 **	0.46 **	0.30 **	1

aPLs—antiphospholipid antibodies; aCL—anti-cardiolipin antibodies; aβ2GPI—anti-β2 glycoprotein I antibodies; aPE—anti-phosphatidylethanolamine antibodies; aPS—anti-phosphatidylserine antibodies; aPT—anti-prothrombin antibodies; * *p* < 0.05; ** *p* < 0.01.

**Table 11 jpm-13-01549-t011:** The aPLs’ correlations among patients with cancer.

	aCL IgG	aCL IgM	aβ2GPI IgG	aβ2GPI IgM	aPT IgG	aPT IgM	aPE IgG	aPE IgM	aPS IgG	aPS IgM
aCL IgG	1	0.31 *	0.12	0.16	0.20	0.25 *	0.38 * *	0.29 *	0.64 * *	0.19
aCL IgM	0.31 *	1	0.11	0.31 **	−0.09	0.20	0.10	0.40 **	0.27 *	0.73 **
aβ2GPI IgG	0.12	0.11	1	0.53 **	0.31 **	0.41 **	0.54 **	0.44 **	0.17	0.16
aβ2GPI IgM	0.16	0.31 **	0.53 **	1	0.26 *	0.41 **	0.31 *	0.45 **	0.13	0.32 **
aPT IgG	0.20	−0.09	0.31 **	0.26 *	1	0.38 **	0.36 **	0.11	0.13	−0.14
aPT IgM	0.25 *	0.20	0.41 **	0.41 **	0.38 **	1	0.44 **	0.36 **	0.16	0.19
aPE IgG	0.38 **	0.10	0.54 **	0.31 *	0.36 **	0.44 **	1	0.40 **	0.38 **	0.09
aPE IgM	0.29 *	0.40 **	0.44 **	0.45 **	0.11	0.36 **	0.40 **	1	0.30 *	0.45 **
aPS IgG	0.64 **	0.27 *	0.17	0.13	0.13	0.16	0.38 **	0.30 *	1	0.23
aPS IgM	0.19	0.73 **	0.16	0.32 **	−0.14	0.19	0.09	0.45 **	0.23	1

aPLs—antiphospholipid antibodies; aCL—anti-cardiolipin antibodies; aβ2GPI—anti-β2 glycoprotein I antibodies; aPE—anti-phosphatidylethanolamine antibodies; aPS—anti-phosphatidylserine antibodies; aPT—anti-prothrombin antibodies; * *p* < 0.05; ** *p* < 0.01.

## Data Availability

Not applicable.

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
