# Peer review of "Criteria and Non-Criteria Antiphospholipid Antibodies and Cancer in Patients with Involuntary Weight Loss"

_jpm, 2023, doi:10.3390/jpm13111549_

Round 1

Reviewer 1 Report

Comments and Suggestions for Authors

Interesting study. Clinically, it may prove to be very relevant in view of the growing number of cancer patients.

It may turn out in the future that some patients should not receive NOACs only VKAs

Data in need of correction:

1. The incidence of thrombotic events seems to be higher shortly before and after the diagnosis. Increased age, comorbidities, or chemotherapy are found among the risk factors cited in the literaturÄ™.

Please add other risk factors such as type of cancer (pancreatic, biliary, ovarian), cage radiotherapy, blood transfusions, vomiting diarrhea during treatment, central catheter, preparations to improve blood count parameters

2. Table 2. Main diagnoses found in the study population.

Please change the table: either write here only the types of cancer % and make a second table with concomitant diseases or change the top heading: causes of weight loss in patients

3. It is necessary to add about anticoagulants in cancer patients. Now, patients can receive NOACs (please cite the literature below), and in the case of positive antibodies, maybe they would have to receive VKAs instead of NOACs-as in antiphospholipid syndrome

A practical approach to the ESC 2022 cardio-oncology guidelines. Comments by a team of experts: cardiologists and oncologists.

Leszek P, Klotzka A, BartuÅ› S, Burchardt P, Czarnecka AM, DÅ‚ugosz-Danecka M, Gierlotka M, KoseÅ‚a-Paterczyk H, Krawczyk-Ożóg A, Kubiatowski T, Kurzyna M, Maciejczyk A, Mitkowski P, Prejbisz A, Rutkowski P, Sierko E, SterliÅ„ski M, Szmit S, Szwiec M, Tajstra M, TyciÅ„ska A, Witkowski A, Wojakowski W, Cybulska-Stopa B.Kardiol Pol. 2023 Sep 3. doi: 10.33963/v.kp.96840. Online ahead of print.PMID: 37660389 

Author Response

We would like to express our gratitude for the assessment of our manuscript, as well as for the valuable suggestions made in your review notes, and for providing the opportunity of a revision.

Please find below a point-by-point response to your comments.

Point 1: The incidence of thrombotic events seems to be higher shortly before and after the diagnosis. Increased age, comorbidities, or chemotherapy are found among the risk factors cited in the literature.

Please add other risk factors such as type of cancer (pancreatic, biliary, ovarian), cage radiotherapy, blood transfusions, vomiting diarrhea during treatment, central catheter, preparations to improve blood count parameters

Response 1: Thank you very much for your comment. The thrombosis risk factors have now been more widely described (lines 51-58).

Point 2: Table 2. Main diagnoses found in the study population.

Please change the table: either write here only the types of cancer % and make a second table with concomitant

diseases or change the top heading: causes of weight loss in patients

Response 2: We thank the reviewer for the comment. The descriptive title of Table 2 has been changed to “Causes of weight loss in the study population” (line 176).

Point 3: It is necessary to add about anticoagulants in cancer patients. Now, patients can receive NOACs (please cite the literature below), and in the case of positive antibodies, maybe they would have to receive VKAs instead of NOACs-as in antiphospholipid syndrome

A practical approach to the ESC 2022 cardio-oncology guidelines. Comments by a team of experts: cardiologists and oncologists.

Leszek P, Klotzka A, BartuÅ› S, Burchardt P, Czarnecka AM, DÅ‚ugosz-Danecka M, Gierlotka M, KoseÅ‚a-Paterczyk H, Krawczyk-Ożóg A, Kubiatowski T, Kurzyna M, Maciejczyk A, Mitkowski P, Prejbisz A, Rutkowski P, Sierko E, SterliÅ„ski M, Szmit S, Szwiec M, Tajstra M, TyciÅ„ska A, Witkowski A, Wojakowski W, Cybulska-Stopa B.Kardiol Pol. 2023 Sep 3. doi: 10.33963/v.kp.96840. Online ahead of print.PMID: 37660389

Response 3: Thank you very much for your suggestion. Details regarding anticoagulant therapy in cancer patients have been provided and the recommended paper has been cited (lines 85-91).

Reviewer 2 Report

Comments and Suggestions for Authors

The article "Criteria and non-criteria antiphospholipid antibodies and cancer in patients with involuntary weight loss " seeks to understand the relationship, if any, between antiphospholipid antibodies in weight loss, hypothesizing weight loss as a marker of cancer.

The data presented are sparse in both the cancer group and the noncancer group.

Of course, cancer selection should be improved, and I would not consider the non-cancer group, which is a heterogeneous group of patients grouped with stochastic selection based on the criterion of weight loss.

The cancer group should be increased and matched with the disease-free group in order to obtain more information about the selection criteria.

Author Response

We would like to express our gratitude for the assessment of our manuscript, as well as for the valuable suggestions made in your review notes, and for providing the opportunity of a revision.

We have attentively considered your comments and we have done our utmost to address every one of them.

Please find below our response to your comments.

The article "Criteria and non-criteria antiphospholipid antibodies and cancer in patients with involuntary weight loss " seeks to understand the relationship, if any, between antiphospholipid antibodies in weight loss, hypothesizing weight loss as a marker of cancer.

The data presented are sparse in both the cancer group and the noncancer group.

Of course, cancer selection should be improved, and I would not consider the non-cancer group, which is a heterogeneous group of patients grouped with stochastic selection based on the criterion of weight loss.

The cancer group should be increased and matched with the disease-free group in order to obtain more information about the selection criteria.

We thank the reviewer for the highly pertinent feedback provided for our manuscript.

The present study was developed as a branch of a broader research grant, implemented in 2008, aiming to describe the etiology, along with new and effective prediction scores and investigative pathways in patients presenting for involuntary weight loss.

Several studies pertaining to this project were previously published. They evaluated the validity of using different variables, such as alkaline phosphatase level, lactate dehydrogenase level, ferritin level, white blood cell count, albumin level, and age, in predicting cancer in patients with involuntary weight loss (Baicus C. et al, 2012; Baicus C. et al., 2014), as well as the link between involuntary weight loss, cancer, and different serum cytokines – tumor necrosis factor α, interleukin 1β , and interleukin 6 (Baicus C. et al., 2012).

One of the secondary objectives of our research was to evaluate the presence of antiphospholipid antibodies in patients with involuntary weight loss. Therefore, all the patients for whom the presence of antiphospholipid antibodies was searched were analysed separately. Our study population only comprised patients addressing the physician for unintentional weight loss, which represented the main inclusion criterion, and ensured the homogeneity of the group. The non-cancer subgroup was used as a control group.

The authors completely agree with the reviewer regarding the reduced sample of cancer patients, and they admit this as one of the limitations of the present study. However, a wide panel of antiphospholipid antibodies was evaluated, and, despite the small number of cancer patients, statistically significant differences were obtained between the cancer and the non-cancer patients, especially when considering the titres.

A future continuation of this research by increasing the number of enrolled cancer patients is an extremely promising idea and we thank the reviewer very much for the suggestion. Unfortunately, since the study was performed in an Internal Medicine Department, achieving a more significant number of cancer patients can be challenging.

We remain open to any further corrections to the manuscript, and we will happily provide any further data regarding the study population of our research.

Round 2

Reviewer 2 Report

Comments and Suggestions for Authors

The paper can be accepted in the present form